# Analysis and Forecast of Traffic Flow between Urban Functional Areas Based on Ride-Hailing Trajectories

**Zhuhua Liao** [1,2], **Haokai Huang** [1,2,*], **Yijiang Zhao** [1,2], **Yizhi Liu** [1,2] **and Guoqiang Zhang** [3]

1   School of Computer Science and Engineering, Hunan University of Science and Technology, Xiangtan 411100, China
2   Metaverse Innovation Research Institute, Hunan University of Science and Technology, Xiangtan 411100, China
3   School of Information Science and Technology, Hainan Normal University, Haikou 571158, China
*   Correspondence: 20010502011@mail.hnust.edu.cn; Tel.: +86-136-0977-0800

**Abstract:** Urban planning and function layout have important implications for the journeys of a large percentage of commuters, which often make up the majority of daily traffic in many cities. Therefore, the analysis and forecast of traffic flow among urban functional areas are of great significance for detecting urban traffic flow directions and traffic congestion causes, as well as helping commuters plan routes in advance. Existing methods based on ride-hailing trajectories are relatively effective solution schemes, but they often lack in-depth analyses on time and space. In the paper, to explore the rules and trends of traffic flow among functional areas, a new spatiotemporal characteristics analysis and forecast method of traffic flow among functional areas based on urban ride-hailing trajectories is proposed. Firstly, a city is divided into areas based on the actual urban road topology, and all functional areas are generated by using areas of interest (AOI); then, according to the proximity and periodicity of inter-area traffic flow data, the periodic sequence and the adjacent sequence are established, and the topological structure is learned through graph convolutional neural (GCN) networks to extract the spatial correlation of traffic flow among functional areas. Furthermore, we propose an attention-based gated graph convolutional network (AG-GCN) forecast method, which is used to extract the temporal features of traffic flow among functional areas and make predictions. In the experiment, the proposed method is verified by using real urban traffic flow data. The results show that the method can not only mine the traffic flow characteristics among functional areas under different time periods, directions, and distances, but also forecast the spatiotemporal change trend of traffic flow among functional areas in a multi-step manner, and the accuracy of the forecasting results is higher than that of common benchmark methods, reaching 96.82%.

**Keywords:** traffic flow forecast; functional area; ride-hailing  trajectories; spatiotemporal correlation; graph convolution neural network

## 1. Introduction

In an urban area, different functions, such as residential or industrial, have different traffic laws and characteristics. That is, the urban planning and function layout have important implications for the journeys of the large percentage of commuters, which often make up the majority of daily traffic in many cities. Consequently, traffic congestion happens very often, especially during rush hour, which seriously affects people's work and lives in many cities. Therefore, the analysis and forecast of traffic flow among functional areas are considered to be the key to solving problems in traffic management decisions [1] which can more effectively help traffic management departments to detect the urban traffic status, flow direction characteristics, and traffic congestion causes [2] to strengthen urban governance and improve traffic efficiency, and also to help commuters plan routes to effectively avoid urban traffic congestion areas on their way to work. Furthermore, the accurate and real-time forecasting of traffic flow within functional areas allows motor vehicles, non-motor

vehicles, and pedestrians to better coordinate their time and comfortably travel under the present road and environmental conditions. That is, through scientific regional traffic management and control, the contradiction between supply and demand among functional areas can be macro-coordinated, traffic tension can be alleviated, and the resources of roads and their facilities can be fully utilized.

From a regional perspective, most research on regional traffic flow divides urban areas by a regular grid [3]. However, this division method not only destroys the city's geographic information, but also causes traffic flow to demonstrate high levels of randomness and inaccuracy. From a forecasting point of view, deep learning is an emerging prediction method that performs well in extracting nonlinear features of traffic flow. However, general deep learning models tend to consider only one-sided factors, lacking a deep analysis of traffic flow in time and space dimensions, which results in certain limitations. In addition, traffic flow between functional areas is influenced by various complex factors, such as: (1) spatial dimension: there are interactions and influences among different functional areas and (2) temporal dimension: traffic flow among functional areas has periodic and sudden changes. Therefore, the means to effectively forecast the traffic flow status between functional areas while fully retaining the traffic flow information is a problem that needs to be solved in this paper.

As shown in Figure 1, this study adopts a new way of dividing regions by integrating urban road topology and area of interest (AOI), whereby cities are divided into several functional regions that effectively reflect regional traffic flows' geographic information to improve prediction accuracy. To extract spatiotemporal features of inter-functional area traffic flows, we combine graph convolutional network (GCN), gated recurrent unit (GRU), and attention mechanism to propose a method for forecasting inter-functional area traffic flows. The method can forecast the dynamics and trends of traffic flow among different functional areas and improve accuracy.

The main contributions of this work are threefold:

1.  We propose an area division method for dividing functional areas based on the urban road network and AOIs, which retains the geographic information of the urban area and classifies the functions of the area.
2.  We propose an attention-based gated graph convolutional network (AG-GCN) method for traffic flow forecast between functional areas. This method considers the network topology of functional areas and the time periodicity of traffic flow between them. Moreover, it allocates the weights of traffic flow between functional areas through the attention mechanism layer to improve forecasting accuracy.
3.  We propose a spatiotemporal feature extraction method based on the functional area network and multi-fragment sequence. This method effectively extracts more precise rules and trend features of traffic flow between functional areas in terms of time and space, thereby improving the forecasting performance of traffic flow between functional areas.

In our experiments, we use a real functional area network to analyze and forecast the direction and location of traffic outflow between functional areas. In addition, we evaluate our proposed method using a real traffic dataset, which enhances the authenticity and credibility of the forecast results. Furthermore, we compare our method with some classic methods in traffic flow forecasting to demonstrate its superiority.

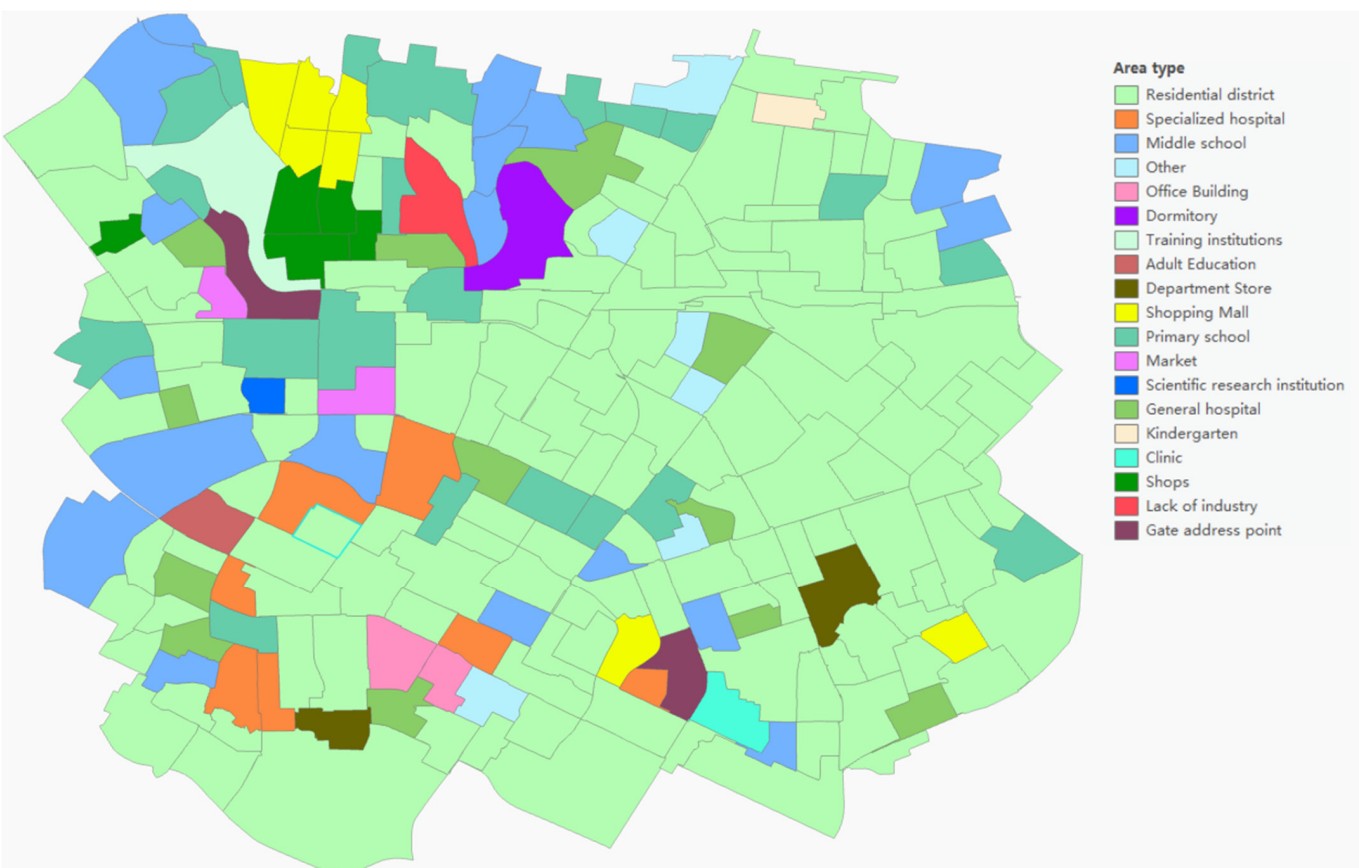

**Figure 1.** Division of urban functional areas (dividing the city into 19 regional types).

## 2. Related Works

In recent years, researchers have explored various approaches to improve the forecast accuracy of traffic flow forecasting [4–7]. These approaches can be broadly classified into traditional methods and deep learning methods. Traditional methods comprise statistical methods and machine learning methods, such as historical average (HA), differential integrated moving average model (ARIMA) [8–10], and vector auto regressive (VAR) [11]. However, these methods have limited accuracy in capturing the complex nonlinear relationships of traffic flow data. Machine learning methods, such as support vector regression (SVR) [12] and the Bayesian model [13–15], can improve forecast accuracy, but they may not be effective in handling complex data in high latitudes.

Due to the complex nonlinear spatiotemporal correlation in traffic flow data, traditional methods may not be able to reveal the deep relationships between various urban spatiotemporal sequence data related to traffic flow. The development of computer vision has led to the emergence of deep learning, which is a suitable choice for traffic flow forecasting tasks. As a result, many researchers have applied deep learning neural networks to traffic flow forecasting and achieved promising results [16,17]. The recurrent neural network (RNN) [18,19] and its variants, such as long short-term memory (LSTM) [20] and gated recurrent unit network (GRU), are frequently used to forecast traffic flow. Although RNN, LSTM, and GRU are effective in capturing the temporal characteristics of traffic flow, they may not capture the spatial characteristics of traffic flow. To address this limitation, scholars have used convolutional neural networks (CNN) [21] to model traffic flow. However, traffic flow data have non-Euclidean structures and nodes with no fixed domain structure, making it challenging to convolve them. Therefore, graph convolutional neural networks (GCN) [22,23] have been introduced, which can effectively mine the spatial dependence of traffic flow. To fully exploit the temporal and spatial characteristics of traffic flow, scholars have proposed T-GCN (temporary graph convolutional network) [24], a

forecast model that considers the temporal and spatial information of traffic flow. This model combines the graph convolutional network (GCN) and gated cycle unit (GRU). GCN is used to learn complex topological structures and obtain spatial correlation, while GRU is used to learn the dynamic changes of traffic flow and obtain temporal correlation.

## 3. Definitions

In this study, our traffic forecast goal is to forecast the traffic flow between functional areas in a certain period according to the historical traffic data. The traditional grid regional division method has lost the integrity of the actual function information of the city. The division based on the administrative region division is not detailed enough and fails to reflect the complete laws and characteristics of traffic flow between functional areas. Therefore, this paper conducts a novel regional traffic flow forecasting based on functional area division.

**Definition 1.** *Functional area: The functional area is divided based on the real urban road network and urban AOI as shown in Figure 2. The urban road network is mainly composed of urban trunk roads and branch roads, which naturally divide the city into (and connect) several small areas. Then, the POI type is identified, and multiple adjacent AOIs of the same type are formed into functional areas. People live and work in functional areas, which are the starting and end points of daily travel. Through functional areas, we can more accurately understand people's travel rules related to functional characteristics.*

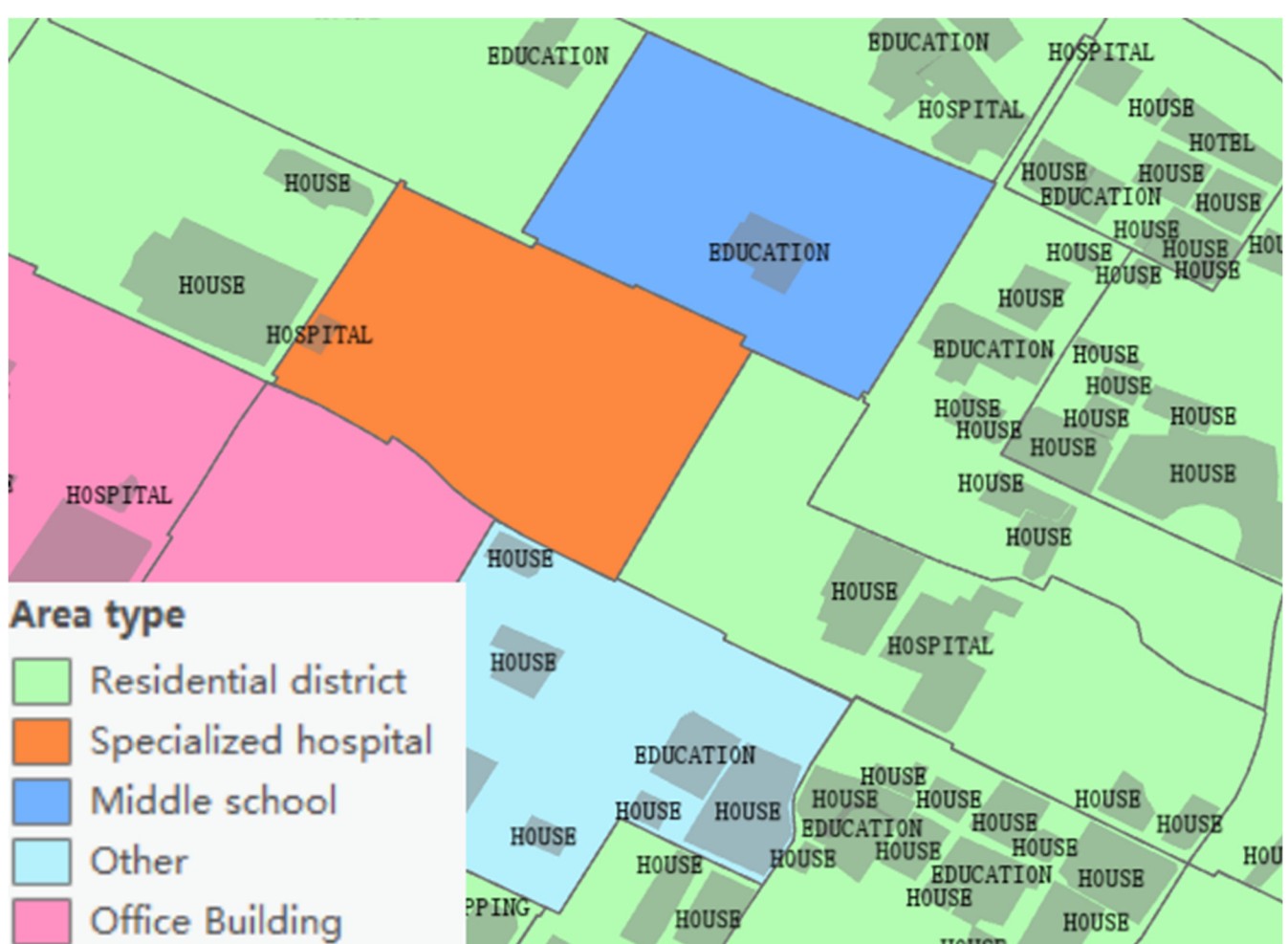

**Figure 2.** Illustration of the functional and connective characteristics of adjacent functional areas.

**Definition 2.** *Area network: In this paper, we use the unweighted graph G = (V, E) to describe the topology of the functional area network. We regard each area as a node and regard a road connecting any two functional areas as an edge, where V is a group of nodes, $V = \{V_1, V_2, \cdots, V_N\}$, N is the number of nodes, and E is a group of edges. Adjacency matrix A is used to represent the connection between areas, $A \epsilon R^{N \times N}$. Adjacency matrix only contains elements of 0 and 1. If there is no connection between two areas, the element is 0; if there is a connection, the element is 1.*

**Definition 3.** *Inter-area flow characteristic matrix, $X^{N \times P}$: Where $X \in R^{N \times P}$ represents the traffic flow matrix and P represents the number of features. The characteristic matrix is utilized to represent the inter-area flow characteristics.*

**Definition 4.** *Inter-area traffic flows: It refers to forecasting the inflow and outflow of traffic flow between functional areas in a future period based on the observed historical traffic flow data of urban areas.*

Let $S$ represent the track set of all vehicles in the continuous time interval $t - 1$, $t$, $t + 1$; then, for the area $g_i$, inflow generated at time $x_t^{in,i}$ and outflow $x_t^{out,i}$ is defined as:

$$x_t^i \begin{cases} x_t^{in,i} = \sum\limits_{T_r \in S} |\{k > 1 | p_{k-1} \notin g_i \wedge p_k \in g_i\}| \\ x_t^{out,i} = \sum\limits_{T_r \in S} |\{k \geq 1 | p_k \notin g_i \wedge p_{k+1} \in g_i\}| \end{cases} \tag{1}$$

where $T_\tau : p_1 \rightarrow p_2 \rightarrow \cdots \rightarrow p_k$ is the trajectory in $S$; $p_k$ is the coordinate of geographical location; $p_k \in g_i$ indicates that the location point is in area $g_i$; $p_k \in g_j$ indicates that the location point is in area $g_j$; $|\cdot|$ represents the cardinality of the set. Therefore, the total traffic flow of a city divided into N areas in time $t$ can be expressed as a tensor $X_t \in R^{2 \times I}$, where $(X_t^i)_0 = X_t^{in,i}$, $(X_t^i)_1 = X_t^{out,i}$.

**Definition 5.** *Inter-area traffic flow forecasts: Given the historical observation value $\{X_t | t = 1, 2, \cdots, n\}$ of the traffic flow between urban functional areas, the regional traffic flow forecast at the next moment is $x_{t+\triangle t}$. Where $\triangle t \in \{1, 2, \cdots\}$ represents the span between the time interval to be forecasted and the current time interval $t$.*

## 4. Methodology

### 4.1. Framework

According to the time-space characteristics of traffic flow between functional areas, an attention based on a gated graph convolutional network (AG-GCN) forecast model is proposed. The architecture of the spatiotemporal forecast model system for traffic flow between functional areas is shown in Figure 3. The preprocessed regional traffic flow is taken as the input, and the periodic sequence and the adjacent sequence are established according to the proximity and periodicity of the inter-area traffic flow. The GCN layer performs convolution operations on the node features using an adjacency matrix and updates the node states with gated functions. The GRU layer concatenates node states from multiple time steps and updates hidden states with gated functions. The attention mechanism layer calculates the correlation between hidden states from different time steps and obtains the final state by weighted summation. Finally, at the output layer, the system maps the final state to the number of vehicles between each functional area in the next time period.

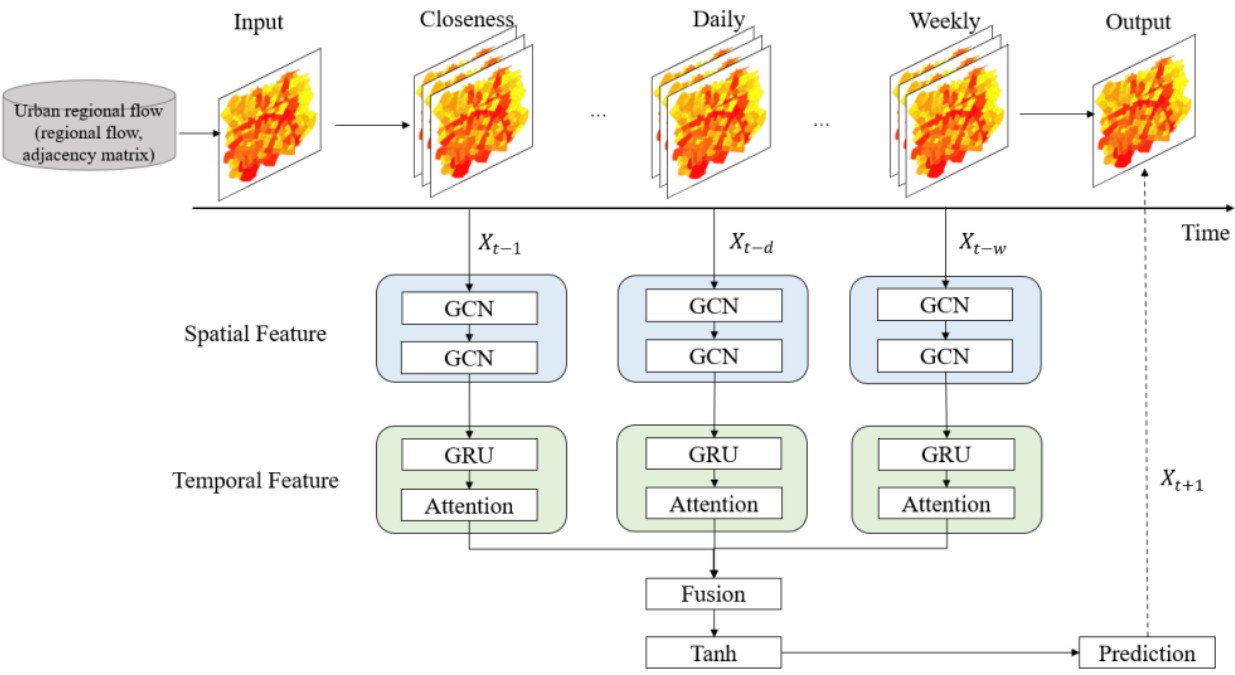

**Figure 3.** Framework of regional traffic flow spatiotemporal prediction model.

### 4.2. Functional Area Division

The functional area division is based on the urban road network and AOI (area of interest). To make the functional area division more orderly and reasonable, 27 kinds of roads are selected from the road network, and urban trunk roads, urban secondary trunk roads, urban branch roads, and internal roads are selected as the urban road network for the division. Areas with a too small an area (such as 0.1 square kilometers) are merged with the surrounding areas. Then, the divided areas are combined with the AOI of the city, and the urban area is classified into several functional areas in 17 categories, including residential areas, shops, hospitals, and primary schools. Finally, massive ride-hailing trajectories, which cover the urban road network, are used to spatially connect the functional areas, and their time series are divided to obtain the inter-area flow characteristic matrix and adjacency matrix.

### 4.3. Spatial Dependency Modeling

The key to solving the problem of inter-area traffic flow forecast is to mine the spatial characteristics of traffic flow between functional areas. A graph convolutional network (GCN) is proposed to solve the topological structure of inter-area traffic flow, improve network efficiency, and better extract spatial features. Its principle is to iteratively update the representation vector of nodes by aggregating the representation vector of their neighbors. Given adjacency matrix A and inter-area flow characteristic matrix X, the GCN model constructs a filter in the Fourier domain. The filter acts on the nodes of the functional area network, captures the spatial characteristics between nodes through its first-order neighborhood, and then establishes the GCN model by superimposing multiple convolution layers.

$$H^{(l+1)} = \sigma\left( \widetilde{D}^{-\frac{1}{2}} \hat{A} \widetilde{D}^{-\frac{1}{2}} H^{(l)} W^{(l)} \right) \tag{2}$$

where, $\sigma$ is the nonlinear activation function of layer $l$, $\widetilde{D}^{-\frac{1}{2}}$ is the root sign after the inversion of the inter-area flow rate matrix $\widetilde{D}$, $\widetilde{D} = \sum_j \widetilde{A}_{ij}$, $i$ and $j$ refer to nodes $v_i$ and nodes $v_j$ of the functional area network, respectively. $\hat{A} = I + A$, $I \epsilon R^{N \times N}$ is the identity matrix, $A \epsilon R^{N \times N}$ is the adjacency matrix of graph G, $\widetilde{D}^{-\frac{1}{2}} \hat{A} \widetilde{D}^{-\frac{1}{2}}$ can be seen as normalizing

the eigenvalues of the degree nodes to reduce the difference between the eigenvalues of different degrees and prevent gradient disappearance or gradient explosion in the later training process, $H^{(l)}$ is the output of layer $l$, and $W^{(l)}$ is the weight parameter of layer $l$.

In this study, a two-layer GCN model is selected to obtain the spatial correlation of inter-area flow, which can be expressed as:

$$f(X, A) = \sigma\left(\tilde{D}^{-\frac{1}{2}} \hat{A} \tilde{D}^{-\frac{1}{2}} \sigma\left(\tilde{D}^{-\frac{1}{2}} \hat{A} \tilde{D}^{-\frac{1}{2}} X W^{(0)}\right) W^{(1)}\right) \tag{3}$$

where $H^{(0)} = X$. $W^{(0)} \in R^{P \times H}$ represents the weight matrix from the input to the hidden layer, $P$ represents the length of the characteristic matrix, $H$ represents the number of hidden cells, $W^{(1)} \in R^{H \times T}$ represents the weight matrix from the hidden layer to the output layer, and $f(X, A)$ represents the output in the forecast area.

In a word, GCN model can obtain the topological relationship between the target area and its surrounding areas, encode the topological structure of the area network and the attributes on the areas, and then obtain the spatial correlation.

*4.4. Temporal Dynamics Modeling*

In the problem of forecasting inter-area traffic flow in urban areas, three types of time-varying characteristics need to be taken into consideration: adjacency, daily periodicity, and weekly periodicity. To fully explore the time characteristics of regional traffic flow, we divide the time series into three parts: the adjacency sequence, the daily periodic sequence, and the weekly periodic sequence.

(1) Adjacency sequence:

$$X_{t-(l_c-1)}, \cdots, X_{t-2}, X_{t-1}$$

(2) Daily periodic sequence:

$$X_{t-d-(l_d-1)}, \cdots, X_{t-d-1}, X_{t-d}$$

(3) Weekly periodic sequence:

$$\left[X_{t-w-(l_w-1)}, \cdots, X_{t-w-1}, X_{t-w}\right]$$

where, $l_c, l_d, l_w$ represents the length of adjacent sequence, daily periodic sequence and weekly periodic sequence of inter-area flow, $d$ represents the number of time intervals included in a day, and $w$ represents the number of time intervals included in a week.

A single cell of GRU neural network has two gates in each hidden neuron unit, namely update gate $z_t$ and reset door $r_t$. The update gate determines whether the information of the previous time period will be transmitted, the reset gate determines how much information will be forgotten, and the remaining information after the reset gate is called the candidate state $\widetilde{h}_t$. After the time data is input, the network will determine the length of the time series according to the length of the input data, and in a time step, all neural units will simultaneously process the data in parallel.

$$r_t = \sigma(W_r x_t + U_r h_{t-1} + b_r) \tag{4}$$

$$z_t = \sigma(W_z x_t + U_z h_{t-1} + b_z) \tag{5}$$

$$\widetilde{h}_t = \tanh(W_h x_t + U_h(r_t * h_{t-1}) + b_h) \tag{6}$$

$$h_t = z_t * \widetilde{h}_t + (1 - z_t) * h_{t-1} \tag{7}$$

where, $x_t$ is the input of current time $t$; $h_{t-1}$ is the hidden layer state at the previous time; $h_t$ is the current output state at time $t$; $r_t$ is the reset gate value; $z_t$ is the update gate value;

$\widetilde{h}_t$ is the candidate status; $W_r$, $W_z$, $W_h$ are the input weight; $b_r$, $b_z$, $b_h$ are the input offset; $\sigma$ is the sigmoid activation function; and $*$ is the hadamard product.

### 4.5. Attention Mechanism

The attention mechanism is a kind of distribution mechanism. According to the importance of an object, resources are redistributed. By reducing the attention to other characteristics of the object, attention is focused on the object where it needs to be concerned to obtain more useful information. Its core idea is to highlight some important characteristics of the object. Traditional attention mechanisms usually place a large amount of weight and attention on adjacent time sequences. However, the time characteristics of inter-area traffic flow are often affected by many external factors, such as holidays, weather, or events. For forecasting future traffic flow, the traffic flow in nonadjacent areas may also be more important than that in adjacent areas. Therefore, based on the work of Wu et al. [25], we use the full connection network to establish the attention mechanism module. Moreover, it automatically learns the weight through the attention mechanism module to improve the forecast accuracy of inter-area traffic flow.

$$A = f(W \circ X + b) \tag{8}$$

$$X' = A \circ X \tag{9}$$

where, $A$ is the weight matrix, $\circ$ is the hadamard product, $X$ is the input, $f$ is the full connection layer, $W$ is the weight matrix of $f$, $b$ is the offset, and $X'$ is the output.

## 5. Results and Analysis

### 5.1. Data Description

The experimental data used in this study was obtained from the online ride-hailing company Didi, specifically, the track data generated by taxis in Chengdu from 1 October 2016 to 31 October 2016, as presented in Table 1. The data was recorded at intervals of 2–4 s and the original data is approximately 50 gigabytes in size. The data covers the second ring road area of Chengdu, spanning longitude 104.042102 E to 104.129076 E, and latitude 30.655191 N to 30.727818 N.

**Table 1.** Trajectory data.

| Driver ID | Order ID | Time (s) | Longitude (°) | Latitude (°) |
|---|---|---|---|---|
| 3a7013bfbbdcb48f7f203ed5d30c8e01 | 464b015cf95322f3c07df5abb908f61f | 1475299381 | 104.05892 | 30.65445 |
| 3a7013bfbbdcb48f7f203ed5d30c8e01 | 464b015cf95322f3c07df5abb908f61f | 1475299399 | 104.0593 | 30.65445 |
| 3a7013bfbbdcb48f7f203ed5d30c8e01 | 464b015cf95322f3c07df5abb908f61f | 1475299421 | 104.06025 | 30.65443 |
| 3a7013bfbbdcb48f7f203ed5d30c8e01 | 464b015cf95322f3c07df5abb908f61f | 1475299418 | 104.06025 | 30.65443 |
| 3a7013bfbbdcb48f7f203ed5d30c8e01 | 464b015cf95322f3c07df5abb908f61f | 1475299406 | 104.05958 | 30.65444 |

The road network data was obtained from the OpenStreetMap (OSM) website. The dataset contains 27 different road types, which have been categorized into eight major types: expressways, urban trunk roads, urban secondary trunk roads, local roads, internal roads, pedestrian walkways, bicycle lanes, and suburban rural roads. This categorization enables a more systematic and structured approach to road classification, which makes it easier to compare and contrast different types of roads and their respective characteristics.

### 5.2. Evaluation Metrics and Baseline Methods for Comparison

#### 5.2.1. Evaluation Index

In order to evaluate the prediction performance of the model, root mean square error (*RMSE*), mean absolute error (*MAE*), coefficient of determination ($R^2$), and accuracy are taken as evaluation indicators in this paper and are defined as follows:

(1) Root mean square error (*RMSE*):

$$RMSE = \sqrt{\frac{1}{N}\sum_{i=1}^{N}(y_i - \hat{y}_i)^2} \tag{10}$$

(2) Mean absolute error (*MAE*):

$$MAE = \frac{1}{N}\sum_{i=1}^{N}|y_i - \hat{y}_i| \tag{11}$$

(3) Determination coefficient ($R^2$):

$$R^2 = 1 - \frac{\sum_{i=1}^{N}(y_i - \hat{y}_i)}{\sum_{i=1}^{N}(y_i - \overline{y_i})} \tag{12}$$

where, $y_i$ is the real value of inter-area traffic flow, $\hat{y}_i$ is the prediction result of inter-area traffic flow, and $N$ is the number of samples. *RMSE* and *MAE* are used to measure the prediction error. The higher the *RMSE* and *MAE* values, the higher the prediction accuracy of the model. $R^2$ calculates the correlation coefficient. The larger the value, the better the prediction effect.

### 5.2.2. Benchmarking Methods

To illustrate the advantages of the proposed AG-GCN model in forecasting traffic flow between functional areas, especially in the training process, we compare the performance of the AG-GCN model with the following baseline methods:

(1)  Historical average (HA) model, which uses the average traffic information of the historical period as a forecast;
(2)  Support vector regression (SVR) model [26], which is a supervised learning algorithm, is often used for time series prediction and has excellent generalization ability;
(3)  Graph convolutional network (GCN) model, which is a neural network architecture that operates on graph data;
(4)  Gated cyclic unit (GRU) model [27], which is a deep learning model based on cyclic neural network (RNN) [28] variants;
(5)  Time graph convolutional network (T-GCN), which is a combination of graph convolutional network (GCN) and gated recursive unit (GRU) that can capture spatiotemporal characteristics and learn change trends.

### 5.3. Experimental Setup

This experiment is based on the open source deep learning library Keras library under TensorFlow to build the deep learning experiment model. All experiments are run on a stand-alone PC. The time slice of the traffic flow dataset in the functional area is set as 10 min. The first 70% of the dataset is used as the model training set, and the remaining 30% is used as the test set. At the same time, the min–max function is used to standardize the traffic flow sequence to the [0, 1] interval. The time slice length is set to 12, the model training step is 2000, the learning rate parameter is set to 0.001, the batch size is 32, and the hidden unit is set to 64. The model uses the Adam function as the model optimizer to optimize the model.

### 5.4. Experimental Result Analysis

In this section, the experimental analysis and evaluation based on a real-world Chengdu dataset show that the AG-GCN model outperforms other benchmark methods in forecasting functional zone traffic flows across different time intervals. This indicates that the proposed model is capable of effectively extracting spatiotemporal features of

functional zone traffic flows and is therefore more suitable for traffic flow prediction and analysis in functional areas. Detailed experimental results are presented in Table 2.

**Table 2.** Performance comparison of AG-GCN model and benchmark models for traffic flow prediction at different time steps.

| Time | Metric | HA | SVR | GCN | GRU | T-GCN | AG-GCN |
|---|---|---|---|---|---|---|---|
| 10 min | *RMSE* | 13.0208 | 9.4440 | 32.3747 | 9.3522 | 9.2249 | **8.7973** |
| | $R^2$ | 0.9334 | 0.9649 | 0.5886 | 0.9656 | 0.9662 | **0.9682** |
| | *MAE* | 8.3185 | 6.1532 | 22.0585 | 6.1413 | 6.0517 | **5.6227** |
| 20 min | *RMSE* | 13.3555 | 9.7939 | 32.6978 | 9.7526 | 9.6493 | **9.1283** |
| | $R^2$ | 0.9302 | 0.9625 | 0.5822 | 0.9628 | 0.9632 | **0.9658** |
| | *MAE* | 8.5104 | 6.3675 | 22.3522 | 6.4781 | 6.3464 | **5.8562** |
| 30 min | *RMSE* | 13.6886 | 10.1409 | 33.5245 | 10.1730 | 10.0547 | **9.2414** |
| | $R^2$ | 0.9270 | 0.9600 | 0.5651 | 0.9597 | 0.9598 | **0.9650** |
| | *MAE* | 8.6976 | 6.5857 | 23.7464 | 6.7547 | 6.6394 | **5.9869** |
| 60 min | *RMSE* | 15.0062 | 11.4120 | 34.1213 | 11.4781 | 11.2326 | **9.4452** |
| | $R^2$ | 0.9133 | 0.9498 | 0.5532 | 0.9493 | 0.9497 | **0.9635** |
| | *MAE* | 9.3891 | 7.3613 | 23.3953 | 7.6925 | 7.4977 | **6.1547** |

(1) Compared to traditional learning methods, deep learning approaches are better suited for handling complex time-series data and extracting features, which can lead to an improved accuracy in forecasting functional area traffic flows. As shown in Table 2, both AG-GCN and T-GCN models, which consider both temporal and spatial features, outperform GCN and GRU models that only consider a single factor.

(2) Among the deep learning methods, the AG-GCN model demonstrates a stronger predictive power than the T-GCN model. This is because the AG-GCN model incorporates an attention mechanism that can more effectively capture the spatiotemporal features of functional area traffic flows and reduce prediction errors. For instance, in forecasting functional area traffic flows 10 min ahead, the AG-GCN model reduces the root mean square error (*RMSE*) by 4.63% compared to the T-GCN model.

(3) Moreover, the AG-GCN model is more appropriate for long-term predictions. While the predictive ability of all models declines as the prediction time step increases due to error accumulation, the AG-GCN model maintains the lowest *RMSE* and mean absolute error (*MAE*) at different time steps (10 min, 20 min, 30 min, or 60 min). This suggests that the AG-GCN model can achieve multi-step predictions of functional area traffic flows.

(4) The AG-GCN model outperformed other benchmark models in forecasting functional area traffic flows based on real datasets. Figure 4 shows the inflow prediction of different models within a day (with 10 min time steps). It can be seen from the figure that the AG-GCN model maintained good and stable prediction performance for 10, 20, 30, and 60 min, while other benchmark models showed a decline of about 1% in prediction performance with increasing time steps. This indicates that the AG-GCN model can learn the complex patterns of functional area traffic flows and capture its spatiotemporal variations.

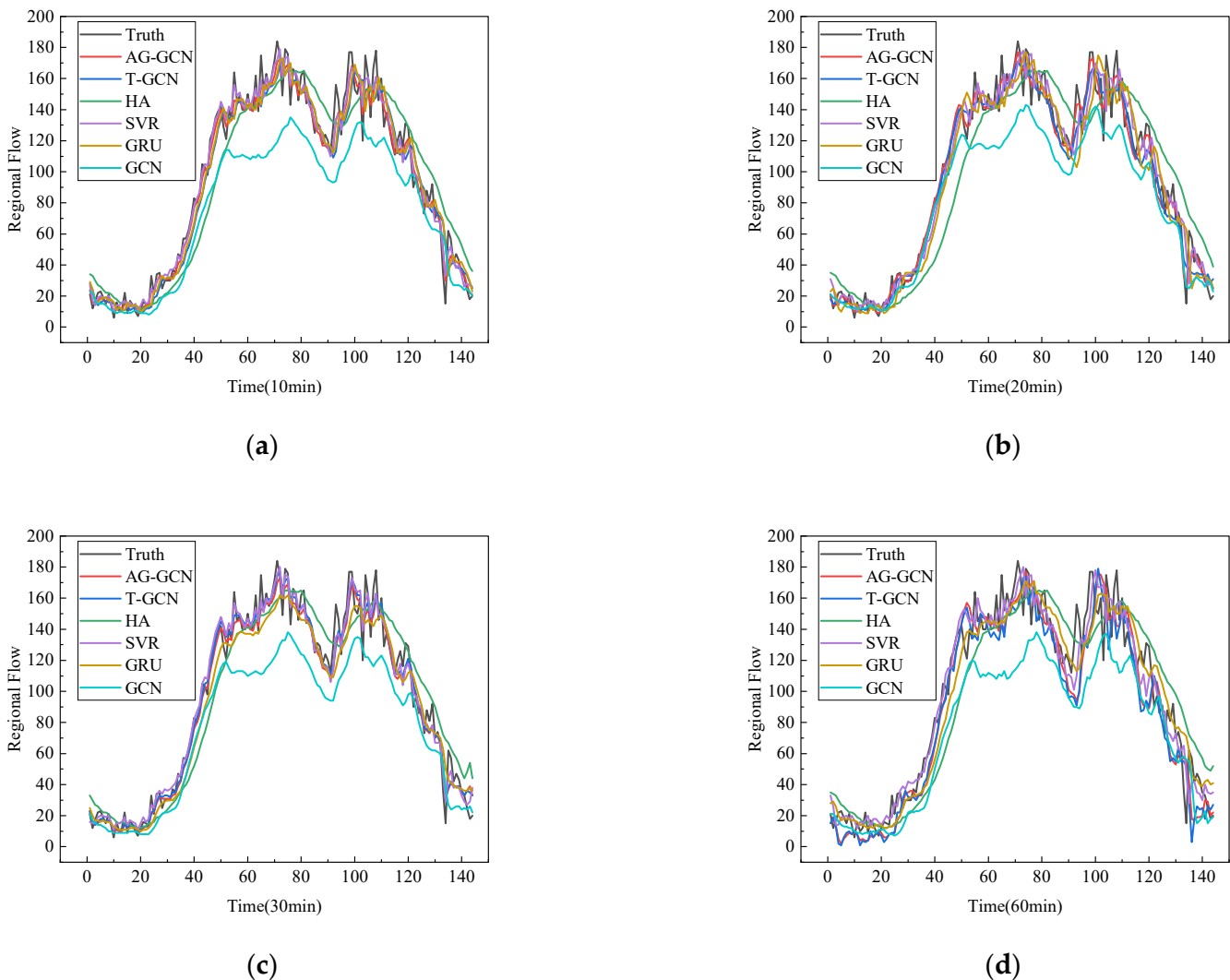

**Figure 4.** Inflow forecast results of functional areas at different time steps (one day): (**a**) forecast 10 min later; (**b**) forecast 20 min later; (**c**) forecast 30 min later; and (**d**) forecast 60 min later.

*5.5. Visualization*

5.5.1. Heat Map

This study uses traffic flow data within different regions to create heat maps that illustrate the traffic conditions in different city areas at various times. Figure 5 shows the heat maps of traffic flow at 6:00 a.m., 9:00 a.m., 6:00 p.m., and 9:00 p.m. The heat maps clearly show that the central area consistently has higher traffic flow than other areas. By analyzing the heat maps of traffic conditions within and between functional areas at different times, we can understand the traffic pressure within different functional areas at different times. Based on this information, we can develop measures to alleviate congestion in advance and reduce urban traffic congestion. This research finding is of great significance for urban transportation planning and management.

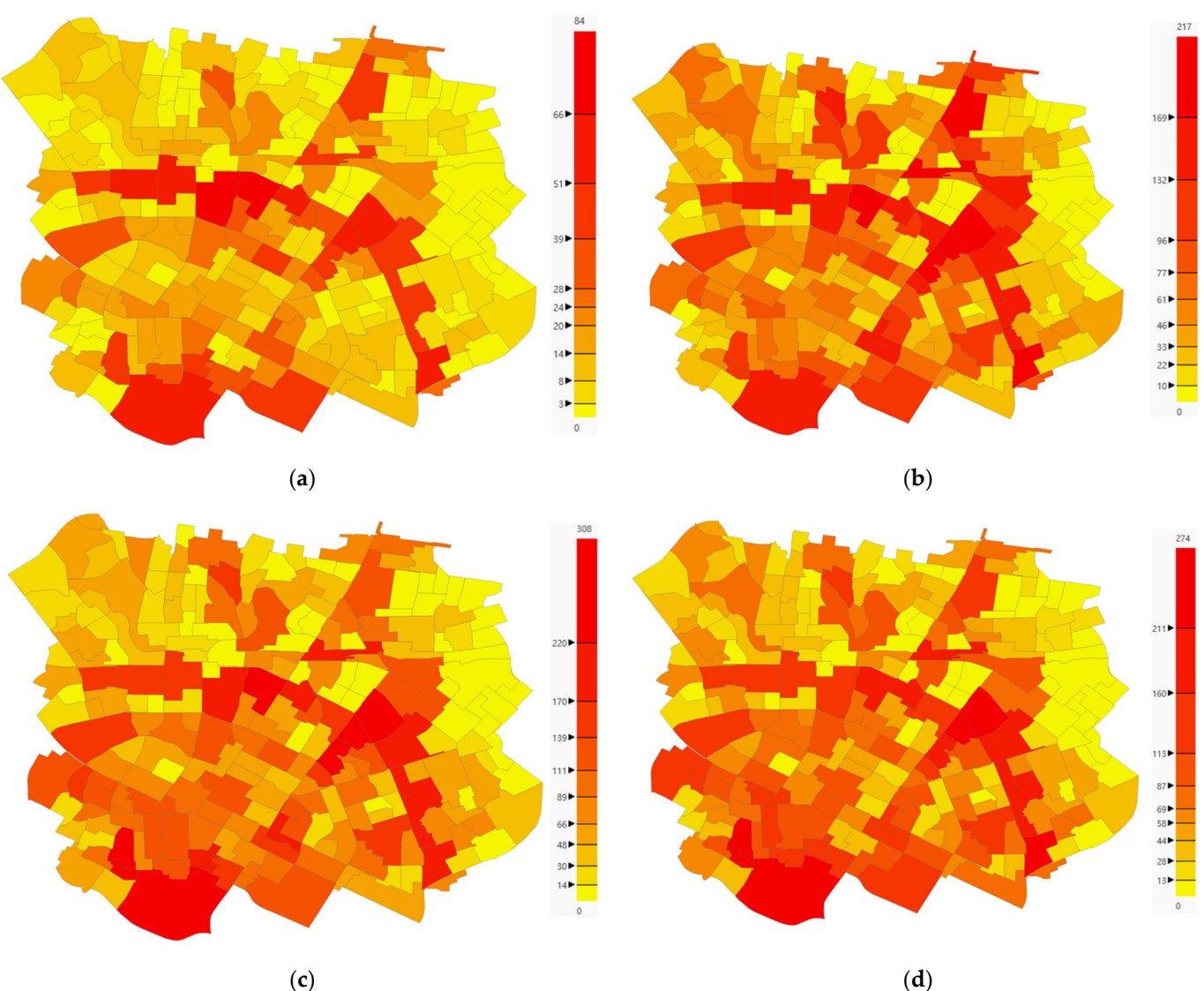

**Figure 5.** Inflow and outflow heat map of functional areas at different time periods: (**a**) 6:00 a.m.; (**b**) 9:00 a.m.; (**c**) 6:00 p.m.; and (**d**) 9:00 p.m.

### 5.5.2. Temporal and Spatial Trend Analysis

How to effectively grasp the traffic connection characteristics between different functional areas of a city is a crucial prerequisite for optimizing the urban traffic dispatching system, alleviating urban congestion, and improving urban operation efficiency. Based on vehicle travel path data, this paper draws the outflow direction maps of each functional area to other areas in order to analyze the traffic connection intensity and directionality between different functional areas. For instance, taking the second functional area by drawing its outflow direction map to other areas during the morning peak period (8:00 a.m.–10:00 a.m.), which is shown in Figure 6, this paper reveals that the traffic flow within this functional area shows a clear concentration tendency and directionality in space. This finding indicates that this functional area is an important traffic hub and has a core position in the overall urban traffic network. In addition, this paper also analyzes the traffic connection intensity and directionality between other functional areas, providing important references for the optimal design of an urban traffic dispatching system.

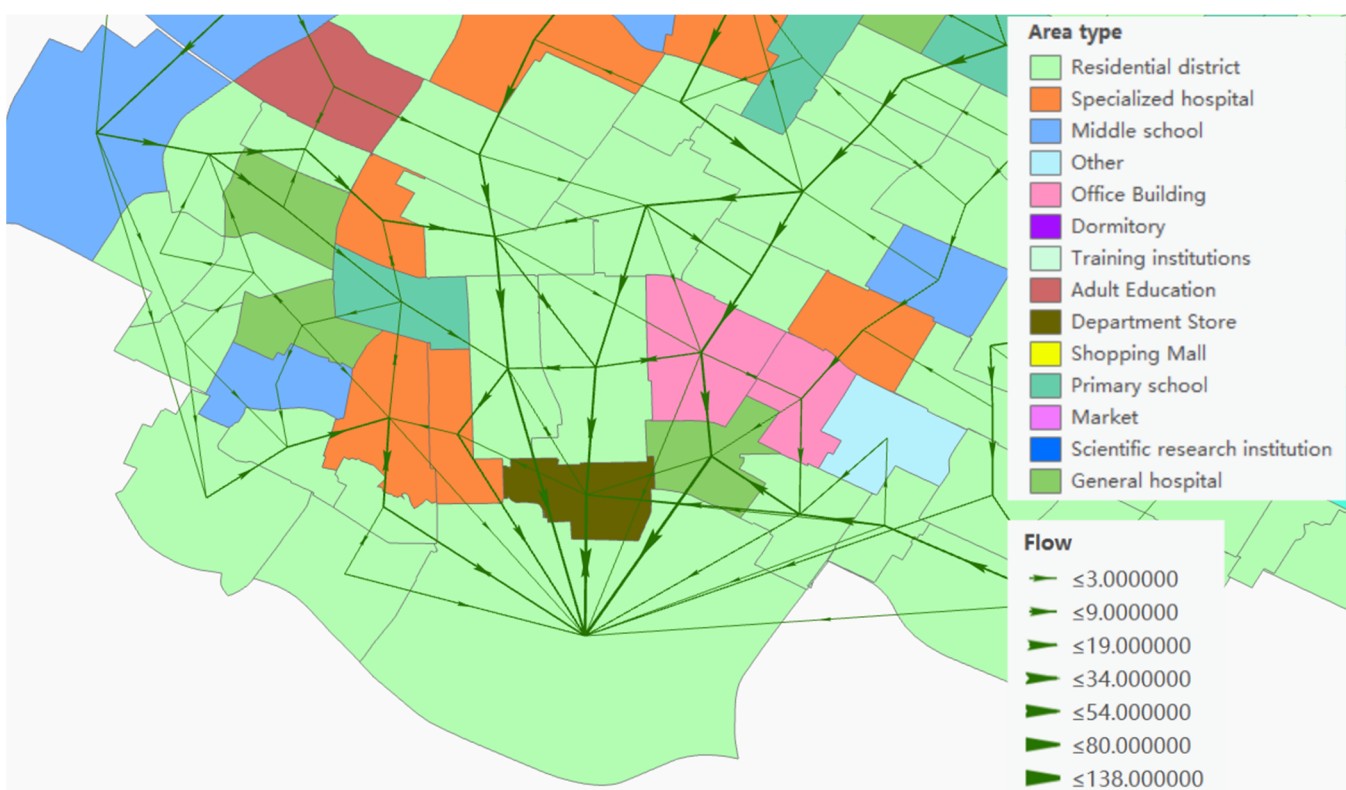

**Figure 6.** Outflow of single functional area to the whole city from 8:00 to 10:00 in the morning.

5.5.3. Traffic Flow between Functional Areas

This study examines the traffic flow characteristics between different functional areas and offers reference for urban traffic management. Figure 7 illustrates the inflow and outflow of traffic flow between each functional zone and its adjacent areas during the morning peak hours (8:00 a.m.–10:00 a.m.), where the arrow direction indicates the direction of traffic movement, the number indicates the size of traffic flow, and different colors represent different functional areas. The following observations can be made from Figure 7:

(1)　There are significant differences in the inflow and outflow of traffic between different functional areas, for example, the traffic volume between commercial zones and residential zones is much larger than that between hospitals and schools;

(2)　There are certain patterns in inflow and outflow of traffic between different functional areas, for example, the peak value of traffic volume between commercial zones and residential zones is reached during the morning peak hours;

(3)　There are the balance or imbalance of supply and demand phenomena between each functional zone and its adjacent areas, for example, commercial zones have a strong attraction to residential zones, while hospitals have a weak influence on schools.

Based on these findings, this study suggests to coordinate the supply–demand relationship between different functional areas from a macro-perspective, to adopt reasonable and effective traffic management measures to alleviate the traffic congestion problem between different functional areas, and to reduce the cost of urban traffic governance.

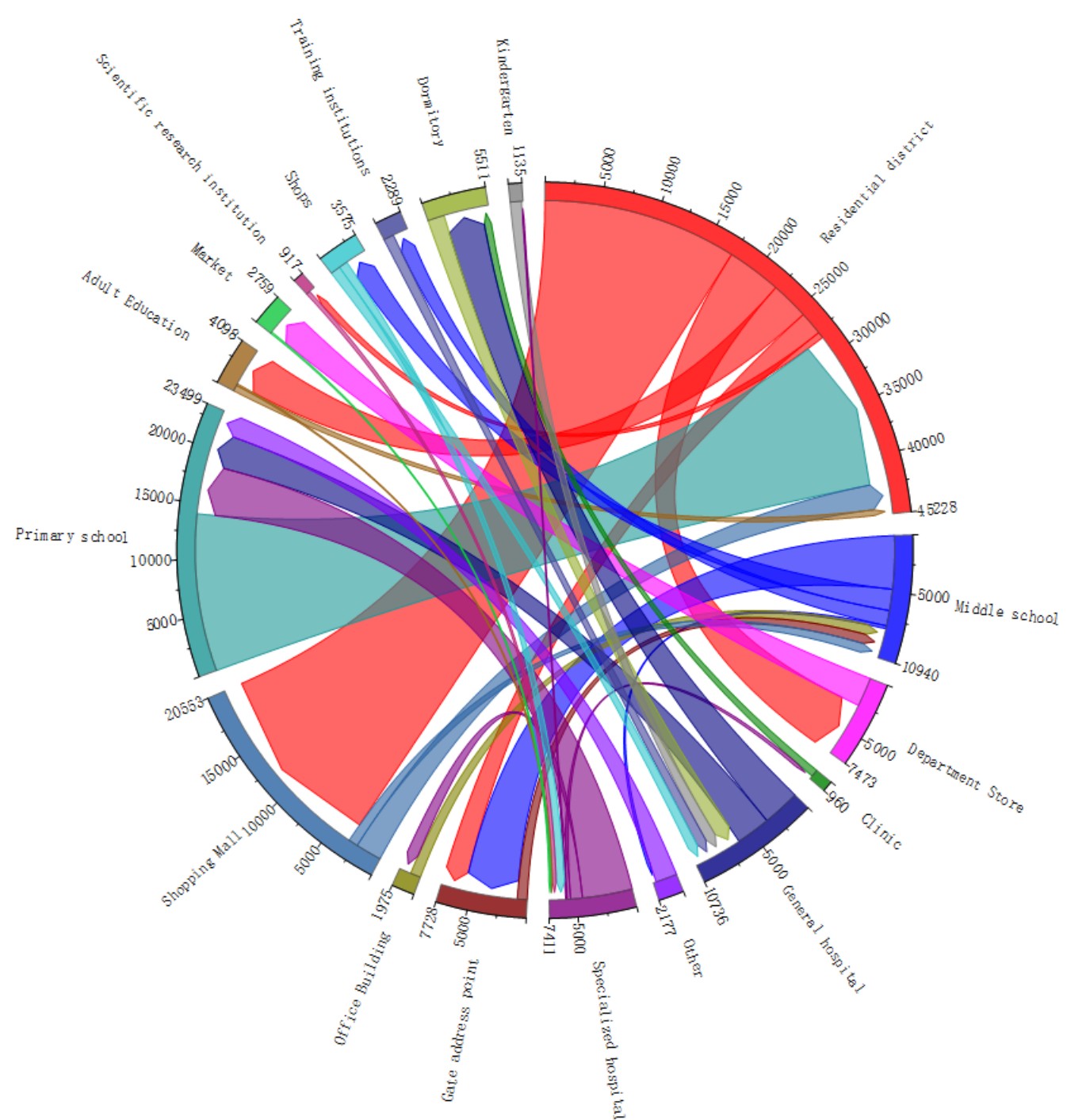

**Figure 7.** Traffic flow of adjacent functional areas in the city.

## 6. Conclusions

In this paper, a novel analysis and forecasting method for the temporal and spatial characteristics of traffic flow between functional areas are proposed. Our method realizes the analysis of the rules, correlations, and scales of traffic flow between functional areas, and can learn the complex patterns and the spatiotemporal changes of traffic flow in functional areas. These will help to solve problems such as the estimation of urban traffic running status and traffic service levels, and the macro-control and planning of urban traffic. The experimental results show that the forecasting accuracy of our method is 98.2%, which is superior to the commonly used non-graph neural network and bench-mark depth learning

methods. Therefore, our method has better forecast performance and can explore and capture the characteristics of nonlinear related factors related to regional traffic flow.

The main work in the future will further research on the laws of traffic flow for different types and distributions of functional areas, and analyze the characteristic relationship of traffic flow between non-adjacent areas to adapt to other more complex traffic flow forecast situations. These will help to improve the accuracy and efficiency of urban transportation management, further promoting the development of urban transportation and a sustainable urban development. In addition, to improve the forecasting accuracy and model generalization performance, future research can utilize multiple sources of data (i.e., buses, private cars, shared bikes, walking) and other relevant data (i.e., meteorological data, population data).

**Author Contributions:** Conceptualization, Zhuhua Liao; methodology, Zhuhua Liao and Haokai Huang; validation, Haokai Huang and Zhuhua Liao; data curation, Haokai Huang; writing—original draft preparation, Haokai Huang; writing—review and editing, Zhuhua Liao, Yijiang Zhao, Yizhi Liu and Guoqiang Zhang; visualization, Haokai Huang; All authors have read and agreed to the published version of the manuscript.

**Funding:** National Natural Science Foundation of China (Grant No. 41871320); Natural Science Foundation of Hunan Province, China (Grant No. 2021JJ30276); the Key Project of Hunan Provincial Education Department (Grant No. 22A0341); National Natural Science Foundation of China (Grant No. 62262018).

**Data Availability Statement:** The vehicle trajectory datasets are available only after the approval of the company 25th March audit is obtained. The data in this paper are mainly from here: [https://outreach.didichuxing.com/], accessed on 25 March 2022.

**Conflicts of Interest:** The authors declare no conflict of interest.

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
