# Peer review of "Analysis and Forecast of Traffic Flow between Urban Functional Areas Based on Ride-Hailing Trajectories"

_ijgi, doi:10.3390/ijgi12040144_

Round 1

Reviewer 1 Report

Article: ijgi-2216073

Title: Ride-Hailing Trajectories Based Analysis and Forecast of Traffic Flow between Urban Functional Areas

REVIEWER COMMENTS

Summary of research presented in this paper:

This paper presents a model for forecasting the flow of traffic between functional areas in an urban environment. The model makes use of a graph convolution network (GCN) and gated recurrent unit (GRU). The contribution of this paper to existing knowledge is threefold. Firstly, a method of functional area division is proposed. Secondly, an Attention based Gated Graph Convolutional Network (AG-GCN) forecast model of traffic flow between functional areas is developed. Thirdly, a spatiotemporal feature extraction method based on functional area network and multi fragment sequence is proposed. The results achieved, suggest that the proposed method performs relatively well to existing methods.

Comments by section:

1.      Introduction:

-        The introduction provides a good rationale and context for the problem addressed in this paper. However, the use of English language and grammar is very poor. The section must be re-written.

2.      Background and related work:

-        the authors are aware of recent literature (mostly less than 10 years). They make a sufficient critical assessment of the short comings of existing methods regarding spatiotemporal forecasting of traffic flow in urban areas. The sectioning unfortunately suffers from a lot of English language and grammar errors and should be revised.

3.      Definition:

-        this section should be titled “Definitions.”

-        Definition 3: … please clarify if this is a custom definition of a characteristic matrix. If so, avoid ambiguity with the definition of an already existing definition of a characteristic matrix in graph theory literature.

4.      Methodology:

-        Described sufficiently but suffers grammar errors like the rest of the paper.

Such as this sentence below:

“In the problem of urban inter-area flow forecasting, there are three types of time dependent characteristics should be analyzed, namely: adjacency, daily periodicity, and weekly periodicity. Therefore, the time slice is divided into three sequences: adjacent sequence, daily periodic sequence, and weekly periodic sequence.”

-        Equations are well described.

5.      Results and analysis:

-        Diagrams and tables are neat.

-        Grammar errors and sentence construction need to be fixed.

Such as this sentence below:

“There are 27 types of roads. we divide the 27 types of roads into 8 categories, including elevated roads and expressways, urban trunk roads, urban secondary trunk roads, urban branch roads, internal roads, pedestrian roads, bicycle lanes, and suburban rural roads.”

-        The chord diagram, Figure 7, requires more explanation. Do not leave it to the reader to figure out. What do the numbers represent? What does the flow represent?

6.      Conclusions: OK.

Overall comments:

-        The paper has a lot of grammatic errors.

-        The overall writing style makes it difficult to understand. The sentences are too long and in some cases the choice of English words is a bit confusing to the reader.

-        The English needs to be improved.

Overall recommendation:

-        The subject addressed in this manuscript is very interesting and contributes valuable insights to the research community, however, the language is very poor.  

-        The manuscript may be resubmitted after extensive language editing by the author.

Reviewer 2 Report

Comments to the Author

Thanks to the authors for this really interesting paper. I do however have several comments that I think must be rectified prior to publication. 

major corrections: 

Abstract: The authors should consider adding a sentence that highlights the [theoretical and/or practical] contribution of this particular study to the abstract.

You need to add clear aim and objectives. 

I need to see a clear statement for the research problem. 

Literature Review: Authors are advised to make efforts to clearly introduce (i.e. by outlining the themes covered in the section) and conclude (i.e. by outlining the implications of the review for the study, as well as the design of the consequent conceptual framework) the Literature Review

the contributions and novelty of the work/method is limited" you need to add : contribution section, you can specifically explain what your contribution is, and the contributions in the sense described above - how does this work push the boundaries of the field.

some parts in the paper need to add more references ( your paper still need to add references to support) , 

you need to write about more urban planning strategies for mitigating climate change and I suggest to use in somehow in the literature about One of the types different planning dealing with climate impacts such as transport planning policy ( use this reference https://doi.org/10.3390/su142316129 

Check the grammar and the page reference format style, please make sure to follow exactly the journal format. 

Please, make sure to put explanation in the text context for each tables, figure, and charts.

You need to do editing and proofreading for the paper to make it high quality enough . 

At end of your manuscript before the conclusion part you need to add more to reflect your results form the analysis and create a develop framework for your conceptual framework, after you have done the research what is the finds result you want to tell us. 

The Abstract part: Please provide more details, in both quality and quantity of the main findings of this paper.

The Conclusion section can also be extended and drawing the future works trend. In this case, at least one paragraph should be provided for the researchers working in this field.

The conclusion sums up the scope of the manuscript, I would also add what could be possible limitations of this framework in the real-world context and what future research goals do you have to expand upon this work. 

You need to add methodology part to explain your method in details. 

You need to add analysis part and add your results  then compare with literature review ( what are the similarities and 

     What are the difference between your results and other’s           studies) you need to more references.

Reviewer 3 Report

This study proposes new spatio-temporal analysis and forecast method of traffic flow between functional areas based on urban ride-hailing trajectories. The authors addressed a very current and interesting topic. It is a strong study in terms of methodology. However, this study needs some corrections:

- Please clearly clarify the research gaps in your study in comparison to previous related topics in the introduction.

- Also,, pls cite more recent and relevant studies (preferably mdpi journal) as the current number of references are not sufficient and outdated.

- The authors proposed a new method for analysis and forecasting the temporal and spatial characteristics of traffic flow between functional areas. i suggest the authors to verify the outputs of this method with others.

- I suggest authors to summarize the most important findings point by point instead of making it very lengthy.

- pls change " We proposes" to "We propose" page3, line 80.

- pls revise this sentence " Our contributions are mainly in three aspects" page 3 line 75,  add verb to the sentence.

- pls add units to Table 1.

Reviewer 4 Report

The paper ''Ride-Hailing Trajectories Based Analysis and Forecast of Traffic Flow between Urban Functional Areas'' is well conceived and methodologically correct.

In the paper, the authors described a new spatio-temporal characteristic analysis and forecast method of traffic flow between functional areas based on urban ride-hailing trajectories.

The authors proposed an Attention based Gated Graph Convolutional Network forecast model of traffic flow between functional areas, which not only considers the network topology of functional areas and the time periodicity of traffic flow between functional areas, but also allocates the weight of traffic flow between functional areas through the attention mechanism layer to improve the accuracy of forecasting.  

The authors proposed the spatio-temporal feature extraction method based on functional area network and multi fragment sequence, which can effectively extract more precise rules and trend features of traffic flow between functional areas in terms of time and space, and help improve the forecast performance of traffic flow between functional areas.       

The proposed model is verified by using real urban traffic flow data.

Remarks:

It would be nice to present the results of previous research more clearly in tabular form as metadata for a clearer presentation.

It would be nice to make a picture of the algorithm of the realized research.

Recommendation:

The paper can be published after minor correction.

Round 2

Reviewer 2 Report

Thank you for improving your paper, Before your paper is ready for publication I have one suggestion in the Conclusions part: 

Try to add : One of the recent trends in transportation solutions or future research is a use  from this paper as reference: 

https://doi.org/10.1016/j.heliyon.2023.e13977